# Can They Stay or Will They Go? A Cross Sectional Study of Managers’ Attitudes towards Their Senior Employees

**DOI:** 10.3390/ijerph19031057

**Published:** 2022-01-18

**Authors:** Kerstin Nilsson, Emma Nilsson

**Affiliations:** 1Division of Occupational and Environmental Medicine, Lund University, 223 81 Lund, Sweden; emma.nilsson.1672@med.lu.se; 2Department of Public Health, Kristianstad University, 291 88 Kristianstad, Sweden

**Keywords:** employability, ageing, work environment, swAge-model, demography, retirement, work–life balance, social support, discrimination, work ability, older worker, senior worker, extended working life, age management

## Abstract

A larger amount of older people need to participate in working life due to the global demographic change. It is the employer, through the manager, who enables employees to have access to measures in the workplace that facilitate and enable a sustainable extended working life. The aim of this study was to evaluate work life factors associated with managers believing their employees can work versus wanting to work until age 65 or older. This cross-sectional study included 249 managers in the Swedish municipality sector. Logistic regression analysis was used to investigate associations between different univariate estimates and in data modelling using the SwAge-model. The result stated that 79% of managers believed their employees ‘can’ work and 58% of managers believed their employees ‘want to’ work until age 65 or older. Health, physical work environment, skills and competence are associated the strongest to managers believing employees ‘can’ work until age 65 or older. Insufficient social support at work and lacking possibilities for relocations associated the strongest to managers believing employees would not ‘want to’ work until age 65 or older. Though, several countries (especially in Europe) have included in their social policy measures that retirement age be increased after 65, proposing ages approaching 70. When these proposals become laws, through obligation, people will have no choice (if they want to or if they can continue working). However, people’s attitudes to work may be different (especially after the COVID-19 pandemic), and this analysis of the participating managers’ attitudes showed there is a difference between why employees ‘can’ versus ‘want’ to work respectively. Therefore, different strategies may be needed to contribute to employees both being able to and willing to participate in working life until an older age. These findings on managers’ perspectives, regarding whether they believe employees would be able to versus would want to work and the SwAge-model, will hopefully contribute to an increased understanding of organisational actions and measures in the process of creating a sustainable extended working life and to increase senior employees’ employability.

## 1. Introduction

Retirement is a valid and socially acceptable way for an employee to withdraw from working life, e.g., from a physically and mentally demanding work situation, depending on whether their personal financial situation will be sufficient with a pension, whether they do not want to continue working due to the social situation and environment in the workplace, their skills not being utilized in the work tasks or that they do not experience motivation and stimulation in their work tasks [1,2]. However, the proportion of senior citizens is continuously increasing in most of the industrial world [3]. Longevity and lower fertility rates characterise the current demographic trend and result in an increasingly ageing population. A consequence analysis carried out by the Organization for Economic Cooperation and Development (OECD) compared the senior boom to have economic and budgetary implications like the social effects resulting from natural disasters [3]. The increasing senior population entails a larger amount of people in the pension system, and the demographic change will result in an increased old-age dependency ratio when fewer citizens active in the workforce must provide for an increasing number of senior citizens. Retirement ages in many countries are being postponed as a way of adapting to the new demographic distribution. Senior citizens are encouraged to continue working and participating in the labour force for as long as possible [3,4,5,6]. This demographic situation stresses the importance of factors motivating senior employees to experience that they can and want to work until an older age. However, in the workplace, it is the employer, through the manager, who enables employees to have access to measures that facilitate labour force participation. Therefore, the manager’s attitudes towards senior employees are important to investigate in relation to enabling a sustainable longer working life.

Previous studies state that some managers hold negative and stereotypical attitudes towards senior employees, such as having more difficulties in adjusting to change, working slower, being less educated as well as holding negative attitudes towards or being frightened of new technology [7]. Perceiving senior employees as stagnant and as an obstacle to organisational development pushes people out of working life early [8,9,10,11,12]. Managers have a key role in the work organisation and on how to motivate and make measures in the work situation to enable employees to extend their working lives. Previous research states that some managers hold negative attitudes towards their senior employees and identify age discrimination [1,7,13,14,15,16,17]. Furthermore, there is published research identifying nine determinant areas connected to employability and whether individuals work or not [1,2,7,8,9,10,11,12,13,14,15,16,17,18,19,20,21,22,23,24,25,26,27,28,29,30,31,32,33,34,35,36,37,38,39,40,41,42,43,44]. These nine areas are also described in the theoretical swAge-model (sustainable working life for all ages) [1] and are: (1) self-rated health and diagnoses; (2) physical work environment; (3) mental work environment; (4) working hours, work pace, time for recuperation; (5) personal financial situation; (6) personal social environment; (7) work social environment; (8) stimulation and motivation in work tasks; (9) competence development, skills, knowledge. Though, to our knowledge, and after further searches in the databases Scopus and PubMed, there appears to be a lack of scientific, published studies and knowledge of managers’ attitudes and beliefs towards why their employees would be able to versus would want to work in an extended working life.

The hypothesis in the discussion about a longer working life is that everyone should work to an older age, but we want to investigate whether it is possible to detect a difference between what contributes to wanting to work versus being able to work. Working in the public sector has been associated with early withdrawal from working life in previous studies [33,34,35,44]. However, more information is needed on the managers’ attitudes towards determinant factors for their employees’ workforce participation and for a sustainable working life. To better understand how managers can motivate employees and make working life healthier and sustainable until an older age, it is of particular interest to examine whether the managers believe that their employees ‘can’ and ‘want to’ work until an older age, associated with the nine determinant areas.

The aim of this study was to evaluate the main factors associated with whether managers believe their employees want to, versus being able to, work until 65 years of age or older in their workplace.

## 2. Materials and Methods

The research design was quantitative, and by a cross-sectional survey investigating factors associated with whether managers believe their employees would be able to or would want to work in an extended working life. The surveys’ objective was specifically to examine whether municipality managers believed their employees would be able to or would want to work, managers’ attitudes towards their senior employees and factors of interest for work participation and retirement.

### 2.1. Study Population

As previously described, working in the public sector has been associated with early withdrawal from working life in earlier studies [33,34,35,44]. The study population in this study consisted of managers in one of the largest municipalities in Sweden. The subjects were identified by the personnel register of employed and there was a total investigation including all the 456 employed managers in the municipality. Therefore, the study population additionally all had the same employer, i.e., the same municipality employer, and the same policies regarding employment conditions, rehabilitation and retirement [45].

The study design of the data collection was a web survey in 2018. The individuals in the present study were sent a questionnaire through their work electronic mail. After two reminders, 249 individuals answered the questionnaire. This corresponded to a response rate of 54.6% and constituted the final study population.

The gender distribution of survey respondents was 29% male managers and 71% female managers. The age distribution in the study group was 27–65 years of age, with a median age of 50 years. Survey respondents, the final study population, generally corresponded with the gender distribution of the managers in the participating municipality, i.e., 30% male and 70% female managers, aged 27–65 years with a median age of 50 years. Additionally, 96% of managers responding to the survey worked 40 h/week (full-time), and 90% were managers in a full-time position. However, we did not have any figure of the distribution of managers working full-time or part-time in the entire municipality.

### 2.2. The Questionnaire

The questions in the questionnaire have been tested and previously used in other studies [7,14,16,19]. The selected statements were based on previous studies executed by our research group [46,47,48], a literature review [15], and other surveys [49,50,51,52]. The questionnaire was tested in a group of 15 individuals (i.e., managers in our own surroundings and other researchers familiar with questionnaires and qualitative investigations) who responded to the web questionnaire and left comments. They comment both on the questions themselves as cognitive testing and readability of items, and on the process, i.e., how it worked to receive information letters and the questionnaire via e-mail as well as how they experienced responding to the online questionnaire. The evaluation of this pilot process resulted in the reformulation of a few question statements before it was distributed to the study population.

Most studies in this research field examine one or a few areas with significance for a sustainable longer working life and employability. The SwAge model is a theoretical model that intends to take a holistic approach and embrace most of all the different aspects and factors that in working life affect the ability to be able and willing to work. We, therefore, used the SwAge model as a model to examine all the areas of influence for a sustainable working life and employability in our intention to contribute to an increased understanding and knowledge in the research area. Accordingly, the questionnaire’s statements included factors associated with employees work situation subdivided into the nine theoretical themes of the SwAge-model [1], i.e.,: (1) health; (2) physical work environment; (3) mental work environment; (4) working hours, work pace, time for recuperation; (5) the personal financial situation; (6) personal social environment; (7) work social environment; (8) work tasks, stimulation and motivation; (9) competence development, skills, knowledge.

The statements’ response options in the nine areas of the SwAge-model were dichotomised from four to two variables, i.e., from highly agree and partly agree to just ‘agree’, and from partly disagree and highly disagree to just ‘disagree’. In the analysis, we were interested in the managers’ answers to these different statements regarding their employee’s work situations and how these were associated with the outcome of two specific questions. The first question was whether the responding managers thought their employees would want to work until 55–59, 60–64, 65 or 66 years of age or older, and the second question was whether the responding managers thought their employees would be able to work until 55–59, 60–64, 65 or 66 years of age or older. The response options were dichotomised at 65 years of age (i.e., working until <65 years of age and ≥65 years of age respectively).

Furthermore, the questionnaire included questions regarding at what age managers identified their female and male employees as senior employees; in which business area they were managers and how long they believed to be able to work themselves and at what age they wished to retire themselves.

### 2.3. Statistical Analysis

The analyses were conducted by the statistical software program IBM SPSS Statistics 25. Regarding the aims of the study, we used the statistical method of logistic regression analysis to investigate the associations between different factors in the work situation measured by statements and the two outcomes in this study, i.e., whether the managers believe that their employees ‘want to work until 65 years of age or older’ and ‘are able to work until 65 years of age or older’. Logistic regression analysis generated odds ratios (OR), as well as 95% confidence intervals (CI) and p-values, for the statement’s association with the two outcomes. For each of the two outcomes, we used the following analytical strategy:

(1) Analysis of each of the nine areas in the SwAge-model: We started with univariate analysis, i.e., we evaluated the associations for one statement at a time. In the second step, we kept the statement with the lowest *p*-value (if *p* < 0.05) and tentatively included all other statements, one at a time. In the third step, we kept the two statements with the lowest *p*-values (if both *p* < 0.05) and tentatively included the remaining statements, one at a time. This procedure continued for as long as the p-values for all included statements were <0.05.

(2) Analysis including all nine areas in the SwAge-model: The multivariate model for all determinant areas together started by including syntaxes of the selected statements from area (1) and area (2), etc., to build a multivariate model. The statements with *p*-values < 0.05 were kept in the model in the next step, in which we also included the selected statements from the next area. This procedure continued until all nine areas were included in a final model. After that, the discarded statements from the nine areas were tested, one at a time, against the final model, to investigate whether the model was robust once more.

## 3. Results

The managers stated different ages to be the possible general retirement age for their employees, i.e., until what age they believed their employees would be able to work and would want to work in general. The proportion of managers who stated that most of their employees would be ‘able work until 65 years of age or older’ was 77%. The proportion of managers who stated that most of their employees would ‘want to work until 65 years or older’ was 58%.

### 3.1. Statements Associated with Managers Believing That Their Employees in General ‘Cannot Work until 65 Years of Age or Older’

The statements from the nine determinant areas were analysed with logistic regression analysis to identify the association between the statements and managers believing that their employees in general **could**
**not work until 65 years of age or beyond** (Table 1). In the first step, the univariate estimates were analysed. The ones that were statistically significant in the next step of the analysis were included in the multivariate model within the determinant area.

Both statements in the determinant area *Self-rated*
*health and diagnoses* were statistically significant and associated with whether the managers believed that their employees could not work until 65 years of age or older, i.e., “The majority of my employees have some kind of diagnosis or chronic disease”, “The majority of my employees do not seem to experience well-being in their daily life”.

Of the seven statements in the area *physical work environment*, only two statements proved to be statistically significant and included in the multivariate model of the area, i.e., ”I experience that my employees in general have many tasks involving a physically demanding work load and heavy lifting” and “I do not experience that my employees have a reasonable physical work load”.

The area *the*
*personal financial situation* consists of only one statement that proved to be statistically significant to whether the managers believed that their employees could not work until 65 years of age or older, i.e., “I experience that my employees in general feel pressured by their personal financial situation”.

In the area *work social environment,* two of seven statements proved to be statistically significant, and accordingly included in the multivariate model, i.e., “My employees in general do not receive sufficient support from me to be able to work until the normal retirement age”, “I am not satisfied with the extent of support that I offer my employees for them to be able to cope with their work tasks”.

Three of seven statements in the area *stimulation and motivation in work tasks* proved to be statistically significant. However, only one of the statements in this determinant area proved to be statistically significant in the final multivariate model, i.e., ”I do not experience it being possible to adjust work tasks to senior employees in my organisation”.

In the determinant area *competence development, skills, knowledge,* three of seven statements proved to be statistically significant. However, only two statements, “I do not experience that my employees have access to sufficient technical support” and “I do not experience it important to keep senior employees in the organization based on their competence”, proved to be statistically significant and included in the final multivariate modelling.

There were no statistically significant associations in the areas *Mental work environment*, *Working hours, work pace and time for recuperation* and *Personal social environment* with whether the managers believed that their employees would not be able to work until 65 years of age or older.

The final step of the analysis, regarding determinant areas associated with whether the managers believed that their employees would be able to work, was a multivariate model of all the determinant areas combined. Five statements proved to be statistically significant and associated with whether the managers believed that their employees would not be able to work until 65 years of age or older (Table 1). The statements belonged to three of the nine determinant areas. One statement belonged to the area *Self-rated health and diagnoses*, two statements belonged to *Physical work environment* and two statements belonged to the area *Competence development, skills, knowledge*. The strongest observed association in the multivariate model was for the statement “I do not experience that my employees have a reasonable physical work load” followed by “I experience that my employees in general have many tasks involving a physically demanding work load and heavy lifting”, “I do not experience it being important to keep senior employees in the organisation based on their competence”, “I do not experience that my employees have access to sufficient technical support”, and “The majority of my employees have some kind of diagnosis or chronic disease”.

### 3.2. Statements Associated with Whether Managers Believed That Their Employees in General ‘Do Not Want to Work until 65 Years of Age or Older’

To identify the association between the statements from the nine determinant areas and whether the managers believed their employees in general **would not want to work until 65 years of age or older**, the association was analysed through logistic regression analysis following the same steps as described above (Table 2). In the determinant area *Self-rated health and*
*diagnoses,* one of the two statements proved to be statistically significant, i.e., “The majority of my employees have some kind of diagnosis or chronic disease”, and included in the final multivariate model in the area.

In the determinant area *Physical work environment,* one of seven statements proved to be statistically significant, i.e., “In my experience it is hard to find work tasks to relocate employees who experience their work environment as too physically demanding”, and included in the final multivariate model.

In the determinant area *Mental work environment*, i.e., the impact on the individual’s mental health caused by the work environment, one of six statements proved to be statistically significant and associated with whether the managers believed that their employees would not want to work until 65 years of age or older, i.e., “I do not experience that there is a general balance between the demands put on my employees in their work, and the control they have in executing their work tasks”.

In the determinant area *Working hours, work pace and time for recuperation,* two of four statements proved to be statistically significant to whether the managers believed that their employees would not want to work until 65 years of age or older. However, only the statement “I do not experience that my employees in general have sufficient possibilities to take breaks when working” proved to be statistically significant and included in the modelling of the final multivariate model for the determinant area.

In the determinant area *Work social environment*, one of seven statements proved to be statistically significant and associated with whether the managers believed that their employees would not want to work until 65 years of age or older, and therefore, included in the multivariate model, i.e., “My employees in general do not receive sufficient support from me to be able to work until ordinary retirement age”.

One of four statements belonging to the determinant area *Stimulation and motivation in work tasks* proved to be statistically significant and associated with whether the managers believed their employees would not want to work until 65 years of age or beyond, i.e., ”I do not experience it being possible to adjust work tasks to senior employees in my organisation”.

In the determinant area *Competence development, skills, knowledge* one of seven statements proved to be statistically significant, i.e., “I do not experience that my employees have access to sufficient technical support”, and included in the final model after the multivariate modelling.

No statements belonging to the determinant areas *Personal financial situation* and *Personal social environment* showed statistically significant associations with whether the managers believed their employees would not want to work until 65 years of age or older.

In the final multivariate model of all determinant areas combined, two statements proved to be statistically significant and associated with whether the managers believed that their employees would not want to work until 65 years of age or older (Table 2). The statements belonged to two of the nine determinant areas. One statement belonged to the determinant area *Work social environment*, and one statement belonged to the determinant area *Stimulation and motivation in work tasks*. The statement included in the multivariate model that proved the strongest observed association to whether the managers believed their employees would not want to work in an extended working life was “My employees in general do not receive sufficient support from me to be able to work until ordinary retirement age”, followed by ”In my experience it is hard to find work tasks to relocate employees who experience their work environment as too physically demanding”.

## 4. Discussion

In a workplace, it is the employer who, through the manager, enables employees to have access to measures that facilitate labour force participation. Therefore, the manager’s attitudes are important to investigate in relation to the facilitation of an extended, sustainable working life. There are some areas that determine whether individuals can versus want to work respectively. Furthermore, these determinant areas are important to the individual’s retirement and retirement planning [1,2,14,15,16,19]. In an organisation or enterprise, it is mainly the manager who makes decisions regarding these determinant areas. The investigation analysis regarded which determinant area was the most important to the managers’ belief that their employees can versus want to work in an extended working life. The multivariate model stated that three areas were statistically significant and associated with managers believing that their employees **could not work until 65 years of age or older**, i.e., *Self-rated*
*health and diagnoses*, *Physical*
*work environment* and *Competence development**, skills, knowledge*. Furthermore, the results of the multivariate model stated that two of these areas were statistically significant and associated with managers believing that their employees would **not want to work until 65 years of age or older,** i.e., *Work*
*social environment* and *Stimulation and motivation in work tasks*. In order to analyse determinant areas of work life participation in this study, the swAge-model was used to structure the analysis. The swAge- model includes nine areas determinant to participation in working life (see the Introduction paragraph). The discussion below follows the structure of the swAge-model and investigates the nine determinant areas associated with the managers believing that their senior employees can versus want to work respectively.

### 4.1. Self-Rated Health and Diagnoses

Individuals’ health situation, i.e., the area *Self-rated health*
*and*
*diagnoses*, is significant to whether individuals can participate in working life at all [1,2,14,15,16,17,18,19,20]. However, in this study, the managers considered the employees who currently participated in working life, and most of the managers (93%) believed their employees to experience sound well-being and health status in their daily lives. Still, the results of the multilevel model in the logistic regression analysis stated that well-being and diagnosed health status among employees were of great importance to whether employees would be able to work until an older age. In the multivariate models, only the employees’ own experience of well-being proved to be statistically significant to both whether employees were able to work as well as whether employees wanted to work until 65 years of age or older. Furthermore, self-rated health and wellbeing being a better predictor for employees’ extended working life than diagnoses have been stated in one earlier study [19]. Therefore, activities and measures to increase employees’ own experience of well-being appear to be important factors in order to increase employability and the possibility of an extended working life.

### 4.2. Physical Work Environment

The results from this study state the physical work environment is of great importance to whether managers believe that their employees would not be able to work in an extended working life. However, the physical work environment did not prove to be statistically significant to whether the manager believed that their employees did not want to work until 65 years of age or older. A poor physical work environment and work conditions increase the risk of work accidents, leave people worn out and push them to leave working life prematurely [1,2,14,15,16,21,22,23]. If so, they would not be able to work and would have difficulties remaining employable. Therefore, taking measures to reduce the risks in the physical work environment appears to be of great importance to employees’ ability to work in an extended working life.

### 4.3. Mental Work Environment

Mental work conditions, stress and lack of control when executing work tasks have also been mentioned as important predictors for employees’ sickness absence and retirement planning [1,2,14,15,16,24,25,26]. In these analyses of this survey, it emerged that the managers did not consider the mental work environment to be of any statistically significant importance to whether their employees would not be able to work beyond the age of 65. However, a statistically significant association was proven with whether their employees would want to work until an older age, i.e., the mental work environment determinant “demands and control” at work. However, this did not prove to be statistically significant in the multivariate model for neither being able to nor wanting to work. Since the mental strain from the work environment accounts for a large proportion of sickness absence from work, managers may need to take the mental work environment more seriously.

### 4.4. Working Hours, Work Pace and Time for Recuperation

The determinant area *Working hours, work pace and time for recuperation* proved to be statistically significant and especially related to managers believing their employees would not want to work until an older age in the univariate model in this study. Some studies highlight a moderate work pace and working hours as important to facilitate sufficient time for rest and recuperation, as well as for employability and the mental and physical ability to execute work tasks [1,2,14,15,16,27,28,29]. Furthermore, different work schedules have been stated as a statistically significant, successful tool to increase employees’ possibility of working until an older age. The manager probably needs to consider that senior employees, in general, need more time for rest and recuperation. Therefore, working hours, work pace and sufficient time for recuperation are important factors for the ability to work in an extended working life.

### 4.5. Financial Incentives

The determinant area *Financial incentives* included research regarding whether the risk of poverty keeps employees in the workforce or whether it is possible to quit working with sufficient personal financial well-being [1,2,14,15,16,30,31,32]. However, financial incentives did not prove to be statistically significant to whether the managers believed their employees would be able to or want to work until 65 years of age or beyond in the final models.

### 4.6. Personal Social Environment

The personal social environment and attitudes in surrounding society also influence withdrawal from working life, for example through marital status, whether the life partner is working or whether the senior employee wants to spend more time with relatives and leisure pursuits [1,2,14,15,16,33,34,35]. The determinant area *Personal social environment* did not prove to be statistically significant in this study. Perhaps the managers primarily focus on the work situation and not as much on the fact that employees also have a personal life outside working life that influences whether they can or want to work.

### 4.7. Work Social Environment

What the management and leadership in the organisation are like, whether the attitudes are positive towards and between employees, whether workers are included in the organisation, or whether there is a stereotypical idea of senior employees as stagnant and an encumbrance [1,2,7,14,15,16,36,37]. Actually, the employees not receiving sufficient support from their manager, a statement belonging to the determinant area *Work social environment*, proved to be the most important, i.e., had the highest statistically significant OR, in the multivariate model of all determinant areas associated with whether the managers believed their employees would not want to work in an extended working life. The manager has a very important role and decision power regarding measures, norms and strategies in the workplace and to enable individual employees to work in an extended working life [1,2,14,15,16,26,36,37]. Therefore, it is important that managers’ attitudes towards their senior employees are positive if society wants a larger amount of people to have the possibility to participate in working life until an older age, due to the demographic development where a larger number of senior citizens need to earn a living.

### 4.8. Stimulation and Motivation in Work Tasks

Individuals need activities. According to previous studies, motivating and simulating tasks have an impact on employees and increase their activity, employability and participation in an extended working life [1,2,14,15,16,38,39,40]. The determinant area *Stimulation and motivation in work tasks* proved to be statistically significant to whether the managers believed their employees both would not be able to and would not want to work until 65 years of age or older. However, in the multivariate model, it only proved significant to whether the managers believed their employees would not want to work until 65 years of age or older. This is in line with earlier studies of employees’ own experiences and wishes to work in an extended working life or not [14,16,34]. Managers, organisations and companies need to take steps to enable the experience of motivation and stimulation in work tasks in order to make employees want to participate in working life until an older age.

### 4.9. Competence Development, Skills, Knowledge

Statements in the determinant area *Competence development, skills, knowledge* proved to be statistically significant to managers believing their employees both would be able to and would want to participate in an extended working life in this study. Competence development, skills, knowledge were stated as a high predictor of whether the manager believed their senior employees would be able to work until an older age in the multivariate model. This is consistent with previous studies that state that the level of education, competence and the possibility of developing skills, but also whether employees are able to utilize their skills in their work tasks, to be important factors to extended work life participation [1,2,14,15,16,41,42,43]. Therefore, it is important that managers and the organisation contribute to enabling the development of competence, skills and knowledge for all employees irrespective of age if the employees should remain employable until an older age.

### 4.10. Limitations and Strength of the Study

The study design is cross-sectional and has limitations since it only shows the result from one point in time. However, this study is the baseline measurement in a longitudinal study regarding factors affecting an extended working life. This baseline investigation will be followed up when employees leave working life, as well as with their health and sickness absence during working life. This entire project will provide a good possibility to investigate the effect of the managers’ attitudes and whether they believe that their employees can and want to work respectively, as well as their attitudes towards measures and action proposals with the aim of enabling a sustainable extended working life, when their employees retire from working life.

The Swedish municipality studied has a larger number of female employees, which corresponded with that most of the study population and the respondents were women.

Although the municipal community included in the study was the eighth largest in Sweden with 456 managers identified to participate in the study, a potential weakness was that 46,4% of the managers in the original study population did not participate. However, the participation rate was 54.6%, compared to other studies this was an expected and normal participation rate of surveys. Additionally, the average age was 50.4 with a range of 25–67, therefore, most managers appear to be in the later stage of their career. Perhaps a study population with a larger proportion of younger managers would have given different results. However, on average, in Swedish working life and in the municipal sector, the average age of managers is high, as in most industrial countries, therefore, the study population corresponds to the general composition of the labour force [3,4,5].

A strength of this study is that it is a total sampling including all the managers in the municipality, and all participation managers work in the same municipality and had the same employer, which minimised the risk of different employment conditions, rehabilitation policies and retirement policies [45].

One strength of this study was the possibility to examine differences between determinant areas and whether managers believe that their employees can or want to extend their working life beyond 65 years of age. All nine determinant areas in the analysis are included in the swAge-model and have been identified in previous studies as very important to retirement and retirement planning [1,2,14,15,16], however, all areas have not previously been included in the same study regarding managers’ attitudes towards their employees’ possibility to work in an extended working life. The results of this study strengthen the theories of the swAge-model.

The questionnaire was sent out after a review of the theoretical basis in the area, the swAge-model, and the majority of the statements in the questionnaire have previously been validated and used in previous studies.

Furthermore, to the best of our knowledge, no previous studies have analysed the distinction between whether managers believe their employees ‘can’ and ‘want to’ work, respectively, in an extended working life.

## 5. Conclusions, Theoretical Contribution and Practical Implications

Work is an important part of an individuals’ life. However, in the workplace, it is the employer, through the manager, who enables employees to have access to measures that facilitate labour force participation. Therefore, the manager’s attitudes are important to investigate in relation to enabling a sustainable extended working life. Since several countries (especially in Europe) have included in their social policy measures from government plans through that the retirement age to be increased after 65, proposing ages approaching 70. When these proposals become laws, through obligation, people will have no choice despite if they do not want to or if they cannot continue working! However, the new situation, generated by the COVID-19 pandemic, changed a lot of the perceptions among managers and employees related to the work processes. To force all people to extend their working life without any organisational work environment activities could be problematic. This study and analysis of the participating managers’ attitudes showed there is a difference between why employees ‘can’ versus ‘want’ to work, respectively. Different determinant areas of work life participation are associated with whether managers believe that their employees are able to work versus that their employees want to work until 65 years of age or older. The results of this study show that health, physical work environment, skills and competence were areas of particular importance to managers believing that their employees **can work until 65 years of age or older**. The results also stated that the well-being, work social environment and stimulation and motivation in work tasks were the most important to managers believing that their employees **want to work until age 65 or older**. These results strengthen the theoretical framework of the SwAge-model [1,5] and could hopefully contribute to a better understanding and development of measure activities that need to be conducted to perform a more sustainable extended working life.

Different strategies may be needed to both contribute to the employees being able to and wanting to participate in working life until an older age. To create a more sustainable working life for all ages, as well as to increase the possibilities to extend working life, organisational measures and activities are needed throughout the entire working life. Individuals’ employability until an older age depends on the nine determinant areas included in the four determinants spheres: health impacts of the work environment; the financial situation; social inclusion, relations and participation; execution of work tasks [1,2,15,53]. To increase the sustainability of working life, and to promote work life participation until an older age, managers must know what measures need to be taken for their ageing employees. One tool that managers could use is to investigate the workplace with support from the nine determinants of the SwAge model, to find out what activities and measures are needed, for example using the template available for this task [1,2,53].

The results from this study will hopefully contribute to the understanding of the process of extended working life. Additionally, the study’s contribution of knowledge can hopefully be used in workplace interventions and future research for a sustainable working life until an older age.

## Figures and Tables

**Table 1 ijerph-19-01057-t001:** Distributions regarding whether the managers believe their employees’ ‘can work’ outcome for the statements included in the univariate and final multivariate model of each determinant area and in total for all nine determinant areas. The corresponding odds ratios (OR), significant value (P) and 95% confidence intervals (CI) obtained from logistic regression. ORs indicate the statements’ relation to the managers’ belief whether their employees’ cannot work until 65 years of age or older.

			The Managers’ Belief Whether Their Employees’ Cannot Work until 65 Years of Age or Older								
Determinant Sphere	Determinant Areas	Statement	Univariate Estimates			Multivariate Model in Each Determent Area			Multivariate Model Including All Nine Determent Areas		
			OR	P	95% CI	OR	P	95% CI	OR	P	95% CI
Health impacts of the work environment	Self-rated health and diagnoses	**The majority of my employees do not seem to experience wellbeing in their daily life**	**4.381**	**0.006**	**1.51–12.68**	**4.238**	**0.009**	**1.43–12.57**	**2.725**	**0.033**	**1.086–6.841**
		**The majority of my employees have some kind of diagnosis or chronic disease**	**3.035**	**0.005**	**1.41–6.54**	**3.126**	**0.005**	**1.42–6.89**			
	Physical work environment	**I experience that my employees in general have many tasks involving a physically demanding work load and heavy lifting**	**7.014**	**<0.001**	**3.05–16.15**	**5.366**	**<0.001**	**2.239–12.860**	**4.660**	**0.002**	**1.779–12.209**
		**I do not experience that my employees have a reasonable physical work load**	**7.130**	**<0.001**	**2.65–19.16**	**4.719**	**0.004**	**1.632–13.644**	**4.832**	**0.008**	**1.526–15.361**
		**I experience that my employees in general run the risk of occupational injury and occupational disease based on the physical work environment**	**4.332**	**0.004**	**1.59–11.84**						
		**I experience that my employees in general have many physically unilateral work tasks**	**2.090**	**0.049**	**1.00–4.35**						
		I do not experience that there is sufficient ergonomic support and aids for my employees work	1.632	0.341	0.60–4.48						
		I do not experience that my employees in general are good at using ergonomic support and aids	1.063	0.87	0.51–2.20						
	Mental work environment	I do not experience that there is a general balance between the demands put on my employees in their work, and the control they have in executing their work tasks	1.813	0.115	0.865–3.800						
		I experience that my employees run the risk of being subjected to violence and threats in their work	1.368	0.319	0.739–2.535						
		I experience that my employees in general run the risk of occupational injury and occupational disease based on the mental work environment	1.338	0.404	0.675–2.655						
		I experience that my employees in general are too stressed in their work due to current circumstances in the work place	1.306	0.408	0.694–2.461						
		I experience that my employees in general are too stressed in their work due to political decisions and circumstances in society	1.225	0.507	0.672–2.233						
	Working hours, work pace and time for recuperation	I do not experience that my employees in general have sufficient opportunity of taking breaks when working	2.066	0.059	0.972–4.390	2.066	0.059	0.972–4.390			
		I do not experience that my employees in general have a good work schedule that enables recuperation between work shifts”	2.003	0.118	0.838–4.784						
		I experience that it can be a problem to keep the work activities running due to lack of temp workers when employees are off work	1.276	0.431	0.696–2.341						
		I experience that my employees in general have too many work tasks due to lack of employees	1.239	0.528	0.636–2.415						
Financial incentives	Personal financial situation	**I experience that my employees in general feel pressured by their financial situation (i.e., having difficulty getting by on their salary, health insurance and/or other social security systems)**	**3.269**	**<0.001**	**1.619–6.601**	**3.269**	**<0.001**	**1.619–6.601**			
Social inclusion, relations and participation	Personal social environment	I do not experience that my employees in general have sufficient opportunity of combining work with their leisure activities and social relations in their leisure time.	2.472	0.062	0.956–6.390	2.472	0.062	0.956–6.390			
		I do not experience that my employees in general have sufficient opportunity of combining work with their family situation, partner, children, grandchildren, etc.	1.964	0.205	0.692–5.572						
	Work social environment	**My employees in general do not receive sufficient support from me to be able to work until ordinary retirement age**	**5.451**	**0.005**	**1.656–17.946**	**4.158**	**0.023**	**1.212–14.268**			
		**I am not satisfied with the extent of support that I offer my employees for them to be able to cope with their work tasks**	**2.415**	**0.017**	**1.169–4.989**	**2.290**	**0.035**	**1.060–4.948**			
		I do not experience that my employees in general receive sufficient support to be able to cope with their work tasks from their co-workers, others in the organization and supporting organizations	1.833	0.190	0.740–4.540						
		I am not satisfied with the quality of the support that I offer my employees for them to be able to cope with their work tasks	1.652	0.333	0.598–4.568						
		I experience that my senior employees are subjected to discrimination/disregard by others in the workplace (co-workers, patients, clients, etc.)	2.115	0.488	0.255–17.569						
		I do not experience that my employees in general have reasonable opportunity to participate in decisions regarding work organisation	1.150	0.780	0.433–3.054						
		I do not experience leadership to be crucial for senior employees’ considerations to keep working after 65 years of age	1.016	0.960	0.558–1.849						
Execution of work tasks	Stimulation and motivation in work tasks	**I do not experience it being possible to adjust work tasks to senior employees in my organization**	**3.900**	**<0.001**	**2.053–7.408**	**3.900**	**<0.001**	**2.053–7.408**			
		**In my experience it is hard to find work tasks to relocate employees who experience their work environment as too physically demanding**	**2.621**	**0.004**	**1.36–5.05**						
		**In my experience it is hard to find work tasks to relocate employees who experience their work environment as too mentally demanding**	**2.230**	**0.030**	**1.081–4.600**						
		I do not experience that my employees in general have reasonable opportunity of participation in decisions regarding their work tasks	3.720	0.070	0.899–15.400						
		I do not experience that my employees in general are satisfied in their daily work	2.400	0.086	0.883–6.520						
		I do not experience that my employees in general have work tasks that they experience as stimulating and meaningful	2.302	0.368	0.375–14.138						
	Competence development, skills, knowledge	**I do not experience that my employees have access to sufficient technical support**	**3.468**	**0.003**	**1.547–7.772**	**3.539**	**0.003**	**1.557–8.043**	**2.834**	**0.030**	**1.109–7.244**
		**I do not experience it being important to keep senior employees in the organization based on their competence**	**2.322**	**0.022**	**1.128–4.778**	**2.497**	**0.016**	**1.188–5.25**	**3.572**	**0.002**	**1.603–7960**
		**I do not experience that my senior employees have the right knowledge and experience for their work tasks**	**2.536**	**0.035**	**1.069–6.015**						
		I do not experience that my senior employees in general have opportunity of continuous competence development	2.131	0.132	0.796–5.702						
		I do not experience that my senior employees in general have the knowledge and experience that enable them to find a job in the eventuality of re-organization and changes	1.469	0.220	0.795–2.713						
		I do not experience that my employees in general have work tasks where they feel they can use their skills and knowledge	1.741	0.653	0.155–19.565						
		I do not experience that my employees in general have knowledge and experience that enables them to be reallocated in our organization	1.055	0.860	0.581–1.915						

**Table 2 ijerph-19-01057-t002:** Distributions regarding whether the manager believe their employees’ ‘want to work’ outcome for the statements included in the univariate and final multivariate model of each determinant areas and in total for all nine determinant areas. The corresponding odds ratios (OR), significant value (P) and 95% confidence intervals (CI) obtained from logistic regression. ORs indicate the statements’ relation to whether the managers believe their employees do not want to work until 65 years of age or beyond.

			The Managers Believe Their Employees’ Not Want to Work until 65 Years of Age or Beyond								
Determinant Sphere	Determinant Areas	Statement	Univariate Estimates			Multivariate Model in Each Determent Area			Multivariate Model Including All Nine Determent Areas		
			OR	P	95% CI	OR	P	95% CI	OR	P	95% CI
Health impacts of the work environment	Self-rated health and diagnoses	**The majority of my employees seem not to experience wellbeing in their daily life**	**2.989**	**0.052**	**0.990–9.027**	**2.989**	**0.052**	**0.990–9.027**			
		The majority of my employees have some kind of diagnosis or chronic disease	1.563	0.234	0.750–3.260						
	Physical work environment	I experience that my employees in general run the risk of occupational injury and occupational disease based on the physical work environment	2.661	0.062	0.951–7.448						
		I experience that my employees in general have many tasks involving a physically demanding work load and heavy lifting	1.744	0.168	0.791–3.847						
		I do not experience that my employees in general have a reasonable physical work load	1.601	0.326	0.626–4.093						
		I do not experience that my employees in general are good at using ergonomic support and aids	1.289	0.422	0.694–2.394						
		I experience that my employees in general have many physically unilateral work tasks	1.303	0.445	0.660–2.573						
		I do not experience that there is sufficient ergonomic support and aids for my employees work	1.192	0.709	0.475–2.990						
	Mental work environment	**I do not experience that there is a general balance between the demands put on my employees in their work, and the control they have in executing their work tasks**	**2.054**	**0.037**	**1.043–4.045**	**2.054**	**0.037**	**1.043–4.045**			
		I experience that my employees in general are too stressed in their work due to political decisions and circumstances in society	1.601	0.070	0.963–2.664						
		I experience that my employees in general run the risk of occupational injury and occupational disease based on the mental work environment	1.698	0.083	0.932–3.093						
		I experience that my employees in general are too stressed in their work due to current circumstances in the work place	1.362	0.245	0.809–2.291						
		I experience that my employees run the risk of being subjected to violence and threats in their work	1.112	0.699	0.649–1.905						
	Working hours, work pace and time for recuperation	**I do not experience that my employees in general have sufficient possibilities to take breaks when working**	**2.376**	**0.017**	**1.164–4.851**	**2.376**	**0.017**	**1.164–4.851**			
		**I do not experience that my employees in general have a good work schedule that enables recuperation between work shifts**	**2.493**	**0.032**	**1.081–5.747**						
		I experience that it can be a problem to keep the work activities running due to lack of temp workers when employees are off work	1.157	0.574	0.696–1.922						
		I experience that my employees in general have too many work tasks due to lack of employees	1.037	0.902	0.583–1.845						
Financial incentives	Personal financial situation	I experience that my employees in general feel pressured by their financial situation (i.e., having difficulty getting by on their salary, health insurance and/or other social security systems)	1.140	0.698	0.588–2.213						
Social inclusion, relations and participation	Personal social environment	I do not experience that my employees in general have sufficient opportunity of combining work with their leisure activities and social relations in their leisure time.	2.225	0.093	0.875–5.658						
		I do not experience that my employees in general have sufficient opportunity of combining work with their family situation, partner, children, grandchildren, etc.	2.067	0.155	0.760–5.624						
	Work social environment	**My employees in general do not receive sufficient support from me to be able to work until ordinary retirement age**	**4.582**	**0.025**	**1.208–17.380**	**4.582**	**0.025**	**1.208–17.380**	**4.972**	**0.022**	**1.263–19.574**
		I am not satisfied with the extent of support that I offer my employees for them to be able to cope with their work tasks	1.671	0.136	0.846–3.300						
		I do not experience that my employees in general receive sufficient support to be able to cope with their work tasks from their co-workers, others in the organization and supporting organizations	1.462	0.378	0.629–3.397						
		I am not satisfied with the quality of the support that I offer my employees for them to be able to cope with their work tasks	1.237	0.666	0.470–3.259						
		I experience that my senior employees are subjected to discrimination/disregard by others in the workplace (co-workers, patients, clients, etc.)	1.394	0.644	0.340–5.708						
		I do not experience that my employees in general have reasonable opportunity of participation in decisions regarding work organization	1.460	0.404	0.600–3.557						
		I do not experience leadership to be crucial for senior employees consideration to keep working after 65 years of age	1.417	0.181	0.851–2.359						
Execution of work tasks	Stimulation and motivation in work tasks	**In my experience it is hard to find work tasks to relocate employees who experience their work environment as too physically demanding**	**2.836**	**<0.001**	**1.657–4.855**	**2.836**	**<0.001**	**1.657–4.855**	**2.812**	**<0.001**	**1.621–4.872**
		**I do not experience it being possible to adjust work tasks to senior employees in my organization**	**1.867**	**0.017**	**1.116–3.123**						
		In my experience it is hard to find work tasks to relocate employees who experience their work environment as too mentally demanding	1.721	0.057	0.983–3.013						
		I do not experience that my employees in general have reasonable opportunity of participation in decisions regarding their work tasks	1.204	0.802	0.281–5.158						
		I do not experience that my employees in general are satisfied in their daily work	1.861	0.208	0.708–4.893						
		I do not experience that my employees in general have work tasks that they experience as stimulating and meaningful	2.226	0.386	0.365–13.577						
	Competence development, skills, knowledge	**I do not experience that my employees have access to sufficient technical support**	**2.622**	**0.018**	**1.180–5.828**	**2.622**	**0.018**	**1.180–5.828**			
		I do not experience it being important to keep senior employees in the organization based on their competence	1.588	0.178	0.810–3.112						
		I do not experience that my senior employees have the right knowledge and experience for their work tasks	1.853	0.147	0.805–4.266						
		I do not experience that my senior employees in general have opportunity of continuous competence development	1.584	0.337	0.620–4.048						
		I do not experience that my senior employees in general have the knowledge and experience that enable them to find a job in the eventuality of re-organization and changes	1.128	0.654	0.665–1.914						
		I do not experience that my employees in general have work tasks where they feel they can use their skills and knowledge	2.792	0.404	0.250–31.212						
		I do not experience that my employees in general have knowledge and experience that enables them to be reallocated in our organization	1.040	0.878	0.627–1.726						

## Data Availability

The data used in this study is managed by the authors. To access this data please contact the authors.

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
