# Peer review of "Can They Stay or Will They Go? A Cross Sectional Study of Managers’ Attitudes towards Their Senior Employees"

_ijerph, 2022, doi:10.3390/ijerph19031057_

Round 1
Reviewer 1 Report
Dear Authors,
The main problem of the paper is the lack of positioning of the research study carried out on the time axis. In fact, it is important to mention when the actual research took place. Furthermore, the authors mentioned that the study is based on a baseline investigation (carried out in .........?!?).
Another not very clear issue is related to a pilot test (lines 134-135) performed with 15 people, a test that is not known when it took place (of the 15 people, how many were used in the main study?). What was the basis for choosing the 15 people? As a procedure for refining the questions, did you follow a Delphi-type investigation? How long did this pilot test last?
How long did it take to rewrite/redesign some questions until the actual study took place?
Then, another discussion is related to the novelty aspects that may support the need for its publication. From a methodological point of view, two elements have been used, now considered classic: the use of a questionnaire containing the nine theoretical topics of the swAge model and the logistic regression analysis.
Here, I want to ask the authors to highlight new and useful aspects to potential interested readers, by emphasizing their own contributions (from a methodological point of view), expressed in the paper, to the analyzed field.
Now, a few specific issues. For instance, please take a look on the first part of the paper title. Please try to reformulate the question in a scientific manner!
In the same time please reconsider the keywords (there are too many: 31). Usually, we could consider something between 6 and 10. In the same time, Introduction and the second section, Materials and Methods, contain phrases that in the end quoted too many papers/articles. I will give you and example. In lines 74-76 you inserted a phrase that quoted too many articles ([1,2,7-44]???). Here, and not only here, you have to differentiate the findings of these papers, by underline by few words (or ideas) some features for these researches. This is necessary to prove credibility in highlighting the issues included in these papers.
Also, pay attention to the page numbering since you included pages with the landscape orientation (containing table 1 and table 2).
Finally, I express my curiosity to ask the authors to express/declare the opportunity of this study (and implicitly what is the added value of this paper), since several countries (especially in Europe) have included in their social policy measures (from government plans) through that the retirement age to be increased after 65, proposing ages approaching 70. When these proposals become laws, through obligation, people will have no choice (if they want to or if they can continue working)! For sure the new situation, generated by the COVID-19 pandemic, changed a lot the perceptions among managers and employees related to the work processes!
Author Response
Thank you for your valuable comments on our manuscript
- The main problem of the paper is the lack of positioning of the research study carried out on the time axis. In fact, it is important to mention when the actual research took place. Furthermore, the authors mentioned that the study is based on a baseline investigation (carried out in .........?!?).
Dear reviewer, the investigation was carried out in 2018, due to an increased work burden an additionally changes during Covid-19 pandemic this article has been delayed.
- Another not very clear issue is related to a pilot test (lines 134-135) performed with 15 people, a test that is not known when it took place (of the 15 people, how many were used in the main study?). What was the basis for choosing the 15 people? How long did it take to rewrite/redesign some questions until the actual study took place?
Dear reviewer, this pilot test was also carried out in 2018 and was not included anyone of the main study participations, but managers in our own surroundings and also other researchers familiar with questionnaires and quantitative investigations.
- Then, another discussion is related to the novelty aspects that may support the need for its publication. From a methodological point of view, two elements have been used, now considered classic: the use of a questionnaire containing the nine theoretical topics of the swAge model and the logistic regression analysis. Here, I want to ask the authors to highlight new and useful aspects to potential interested readers, by emphasizing their own contributions (from a methodological point of view), expressed in the paper, to the analyzed field.
Most studies in the field examine one or a few areas with significance for a sustainable longer working life and employability. The SwAge model is a theoretical model that intends to take a holistic approach and embrace most of all the different aspects and factors that in working life affect the ability to be able and willing to work
- Please reconsider the keywords (there are too many: 31). Usually, we could consider something between 6 and 10. In the same time, Introduction and the second section, Materials and Methods, contain phrases that in the end quoted too many papers/articles. I will give you and example. In lines 74-76 you inserted a phrase that quoted too many articles ([1,2,7-44]???). Here, and not only here, you have to differentiate the findings of these papers, by underline by few words (or ideas) some features for these researches. This is necessary to prove credibility in highlighting the issues included in these papers.
We have now fewer keywords, however the study includes a lot of different areas, as the swAge-model. That is also the reason to that the quote includes as many articles.
- Also, pay attention to the page numbering since you included pages with the landscape orientation (containing table 1 and table 2).
The problem with the page numbering is in the journal's template and I can not do anything about it unfortunately. I would therefore like to ask the technical editor to help adjust the page numbering.
- Finally, I express my curiosity to ask the authors to express/declare the opportunity of this study (and implicitly what is the added value of this paper), since several countries (especially in Europe) have included in their social policy measures (from government plans) through that the retirement age to be increased after 65, proposing ages approaching 70. When these proposals become laws, through obligation, people will have no choice (if they want to or if they can continue working)! For sure the new situation, generated by the COVID-19 pandemic, changed a lot the perceptions among managers and employees related to the work processes!
Dear reviewer, thank you for this question. Hopefully this study could contribute to a better understanding of measure that need to be done to perform a sustainable working life for all ages. Text about this is now included in the Conclusion part of the paper.
Reviewer 2 Report
Dear Authors,
Thank you for taking up a very interesting topic of „a cross-sectional study of managers’ attitudes towards their senior employees”. Overall the paper is valuable, however, several issues need improvement, mainly in the area of ​​research methodology description.
The list of issues to be improved is not long and presented below:
1) The abstract should be modified as its structure is inconsistent. Why did the authors separate the sub-headings "methods" and "conclusions"? Following this logic, one should indicate, for example, "objectives", "results", etc. It is worth making the abstract structure more clear.
2) The number of keywords is unacceptable. The norm is to indicate several (5-6) keywords, and the authors listed them over 30.
3) Why were no hypotheses set after the literature review and before starting the study? If the research was only descriptive in nature, it is worth emphasizing.
4) What was the scale in the survey questionnaire?
4) Among the limitations, the authors did not mention the most important one, i.e. the limited representativeness of the research sample due to non-random sampling. This is not a big problem as the response rate was high, but it is worth mentioning.
5) It is not a good idea to combine conclusions, theoretical contributions, and practical implications as one section. I propose to leave contributions and implications as section 5 as is currently the case and provide conclusions in supplementary section 6 (where the results and conclusions of the research will be summarized).
I encourage you to revise a manuscript.
Best regards,
The reviewer.
Author Response
Thank you for your valuable comments on our manuscript
1) The abstract should be modified as its structure is inconsistent. Why did the authors separate the sub-headings "methods" and "conclusions"? Following this logic, one should indicate, for example, "objectives", "results", etc. It is worth making the abstract structure more clear.
Dear reviewer, the abstract is now rewritten due to your point.
2) The number of keywords is unacceptable. The norm is to indicate several (5-6) keywords, and the authors listed them over 30.
We have now fewer keywords, however the study includes a lot of different areas, as the swAge-model, and the keywords are many to reach the different research areas.
3) Why were no hypotheses set after the literature review and before starting the study? If the research was only descriptive in nature, it is worth emphasizing.
The hypothesis in the discussion about a longer working life is that everyone should work to an older age, but we want to investigate whether it is possible to detect a difference between what contributes to wanting to work versus being able to work
4) What was the scale in the survey questionnaire?
Ordinal scale
5) Among the limitations, the authors did not mention the most important one, i.e. the limited representativeness of the research sample due to non-random sampling. This is not a big problem as the response rate was high, but it is worth mentioning.
It was not a random sampling, but a total sampling including all the managers in the municipality.
Round 2
Reviewer 1 Report
Dear Authors,
The comments and, implicitly, the changes made by the authors are insignificant and totally irrelevant (and as such I declare them totally unsatisfactory). The multitude of question marks included in my previous review did not find sufficiently enlightening, topical, and convincing answers. The modified version is very little changed compared to the initially reviewed version.
I found out that the research took place in 2018. Considering the events started worldwide in March 2020 (COVID-19 pandemic) the proposed study is out-of-date and irrelevant in relation to the analyzed topic.
In addition, the authors mentioned aspects based on the information in the dataset that has as baseline survey conducted in 2018, and now the feeling is that variations are being tried on the same theme, without a purpose / gain in itself from a scientific point of view. The authors mentioned already two studies in the Literature review and References list (at least the papers in the reference list from numbers 7 and 25) which had as elements of information the dataset used in this study originated from a baseline survey conducted in 2018. Those data have reaches their utility limit and it is necessary to conduct a new survey in lights of the events of the last 21 months (e.g COVID-19 pandemic)!!! Therefore, my decision is to reject the paper because the research is based on a survey conducted in 2018. The last 21 months have changed the perceptions and attitudes of people, managers and decision makers regarding the working period, the way employees are involved in various fields (health care, IT, administration, education and so on), opinions on the retirement age etc. In fact, this study has non-updated/ old findings! I do not see the relevance of this study in terms of opinions expressed in 2018!
In addition, all these months marked by the COVID-19 pandemic have changed the perceptions of micro and macro decision makers in relation to employees over 65 years of age.
Also, the title of the paper does not have an academic statement and in a scientific sense, having rather a resonance close to a newspaper title. Furthermore, although this was mentioned in the previous review, a very high number of keywords remain (i.e. 23), which makes me wonder if the authors have carefully read the instructions dedicated to the authors, and in addition, if they personally in what they have read and/or written over the years have they found something similar!?
Taking into account all these aspects, I consider the paper to be in no way useful to those interested, because the subject data have changed and the options of policy makers have had to be adapted to the new context!
The issues that need to be highlighted must be put in line with the effects generated during the COVID-19 pandemic!
Author Response
Thank you for the review and suggestions to improve the paper!
We are sorry that we do not follow your but the other reviewer comment regarding how to improve manuscript. The reviewers have quite different comments, and we must go in one direction.
As we can see the greatest issue in your comments is that we do not handle the pandemic. However, the pandemic is not the topic of this paper. You stated in your review report that “all these months marked by the COVID-19 pandemic have changed the perceptions of micro and macro decision makers in relation to employees over 65 years of age”. Maybe you right that it has been changed in some country, however not in Sweden and some other countries and this paper analyse result from a survey in Sweden.
Kind regards!
Reviewer 2 Report
Dear Authors,
Thank you for addressing my comments. In my opinion, the paper can be recommended for publication in its current form.
Best regards,
The reviewer.
Author Response
Thank you fo
Thank you for the review and important suggestions to improve the paper!
Kind regards!
This manuscript is a resubmission of an earlier submission. The following is a list of the peer review reports and author responses from that submission.
Round 1
Reviewer 1 Report
Ref: IJERPH-1280948
Title: Managers’ attitudes and beliefs whether their employees would be able to and would want to work in an extended working life or not, and measure activities to increase their employees’ possibilities of working.
Thank you for asking me to review this manuscript. The topic focusing on aging workers is very important considering the future of work life and our need to support these workers to remain in the workforce with societal, economic and public health impacts. Please find below my comments:
- The title is very long and unwieldy. You tend to get lost in at as a reader. It also doesn’t really make sense at all. I wonder whether something has got lost in translation here.
- Overall comment: In general, the English in this manuscript requires considerable editing if it is to be considered for publication. I would recommend strongly that the authors have their manuscript proofed for English before resubmitting. I have not provided comments on the numerous grammatical areas etc, as they are too numerous. One important note, is that many words are hyphenated?? I am not sure why? For example sched-ule (line 81), as-sociated (line 333, be-yond (line 334), im-portance (line 391), just to give you a few.
- Introduction, Lines 70-76: was this a scoping or systematic review that you conducted if so please state this. Then provide citations for your statements.
- Introduction, Line 91 (and elsewhere): I would avoid using the phrase “this present baseline study” as you are not presenting any data beyond this cross-sectional survey. Just state what this study design is to remove any confusion. In the methods section, you can position this study in the overall context of the larger study.
- Methods, Line 97: I would consider removing the first sentence. The statement “to obtain quantifiable information…. we decided to be quantitative…” is redundant, saying the same thing. Also, quantitative data is not always generalizable, so this statement is not accurate either.
- Methods, Line 100-101: Remove the mention of the analytic approach from here as this is described in section 2.3.
- Methods, Line 109: Your references need to be fixed here. However, I do not feel that this sentence is needed in the methods section as it does not describe the population in this study. I would remove from here, and ensure that this point is made in the introduction.
- Methods, Line 114: Typo: Why is there a 45 at the end of this sentence?
- Methods, Line 117-127: Do you have any indication if the respondents were representative of the full manager sample with respect to their demographic, as detailed here? Please state either way. If you don’t have this data, you can list this as a limitation i.e., we do not know whether our sample was representative of all of the managers in our sample and may have been biased etc…
- Methods, Line 133-134: Can you provide more detail on this pilot process? Was this similar to cognitive testing of the items?
- Methods, Line 144: “Mental Work situation” – Do you mean psychosocial working conditions or cognitive/emotional/psychological job demands? These are quite different constructs.
- Methods, Primary outcomes: What was your rationale/theoretical premise for the two outcomes? I would like to see a section of these, and also some data as to the “testing” of these outcomes, so that the reader can be confident as to their construct/content validity and reliability.
- Methods, Line 142-150: I found this a bit confusing. How were the manager’s responses subdivided into the nine themes and who did that. Or do you mean that the manager’s rated something using these nine different aspects?? If they did how was this asked in the survey? What was the premise for this? This is so unclear, as it is written now, and I am not sure what you are getting at here. These nine aspects need to defined operationally so that it is very clear to the reader how these are defined.
- Methods, General: So I am assuming that besides the two primary outcome questions, there was nothing else asked on the survey?? This seems odd. But nothing else is described in the methods that I can see. Maybe questions about the nine aspects based on swAGE model, but this is unclear. In the statistical analysis section, you vaguely refer to “different factors in the work situation measured by statements” but this is not explained anywhere else in the methods. Also, the results section starts off with the findings from a questions that isn’t even included in the methods section (line 185-198)??
- Methods, Statistical Analysis, Line Lines 168-175: So you used a ?forward step-wise selection of the variables?
- Methods, Statistical Analysis, Line 176: I have no idea what you mean by “Measure variables”? Are these the outcome measures (DVs) measured with the two questons detailed in the methods. These are not described anywhere that I can see.
- Methods statistical analysis: Did you check your final model for goodness-of-fit, i.e. how your model reflects the real data? Did you conduct any other post-estimation diagnostics for your model?
- Methods statistical analysis: What statistical software program did you use to conduct your analyses?
- Results, General: Can you use subheadings to structure your results to make it easier to navigate with respect to the nine areas or bold or underline these for ease of readability?
- Results, Health, diagnoses and function variations: I do not know what is meant by “function variations”?
- Tables. Can the items be reordered with respect to strength of OR from high to low?
22. Discussion, General: Can you use Subheadings to structure your discussion to improve readability?
Reviewer 2 Report
- It is suggested that the authors may revise the title to make it more clear and precise.
- Generally there are 3 to 6 key words, while the authors have included almost 30 key words, this may cause distraction.
- No need to mention Scopus or other database.
- In line 81, work sched-ule?
- In line 109 and 114, reference should be in [ ].
- In section 2.1, the authors argue that the same employer minimize risks of different employment conditions. However, in the following section, there is little explanation of model set up for logistic regression. Specifically, we know little about whether the heterogeneity has been controlled in the model, such as the career. As can be seen in the second paragraph of section 2.1, there are so many different positions. Thus, it is hard to say that some of the polices remain the same as argued by the authors. For example, Care, health care, rehabilitation work can be quite different and may have specific requirements. If such effects are not controlled, the results may not hold.
- The authors choose public sectors as research sample, while the concern is whether it can represent the situation in the industry? For example, there are more female employed as managers in the research sample of the public sector. However, this may not be the case in other sectors.
- In line 185-188, do they really to do so? I believe there are some commonly accepted rules of defining elderly.
- The method the authors used to select variables for multivariate analysis is not that convincing. The statements in each area reflects different dimensions no matter how the univariate test looks like. Besides, why don’t you consider the aggregated score of statements in each area directly? Even though a variable is not significantly associated with the dependent variable, it might become significant in multivariate analysis. A correlation matrix can provide more details about the relations among variables. To sum, the authors may provide some literature to support the use of current method.
- The table should be revised to make it easier to read. The consistency is another issue. For example, if the authors bold the significant variables, they the stars shown in table 3 should be deleted. Or the authors can add stars to all significant variables.
- The authors pointed out that less is known from view of managers. Even though the perspective is different from prior studies, the authors still need to provide the rationality. Why does the perception of managers’ attitudes matter? Why don’t directly check with the employees?
- The average age is 50.4 with a range of 25-67. This showed that most managers are in the later stage of the whole career life. This may lead to less generation gap and share more similarities between managers and employees. However, the younger generations who are on managers’ positions may have different way of thinking and criterion to evaluate workability of the elderly. Thus the author may explain the data with more caution.
Reviewer 3 Report
Some simple problems:
1. Correct in the line 109 the cited studies format! Same in the line 114.
2. The values for OR (in the case of univariate estimates) in the table 1 and table 2 are difficult to be followed by the readers!
Challenging tasks:
- I will start from the paper' subject and title. Of course, the subject could be considered as important and also useful for generate, maybe some info for Public Health and Retirement Policies. The main problem is that "The dataset used in this study originated from a baseline survey conducted in 2018". Why? For sure the last 15 months generate important movements in the mentality of the population. Many thing are happen, many things are changed. For instance, we discovered the teleworking or we have to face with overcrowding and overloading of hospitals. I'm afraid that many of the responses gave in the questionnaire, now, they will be different. We need much more staff in the hospitals, therefore is maybe obvious that for the medical staff the working time will be extended (as mandatory) beyond 65 years. Also, in many other domains such IT, finances, administration this new concept, teleworking will help people to work easier from their home;
- The methods used by the authors is, now, very traditional, very classic;
- The findings coming from the survey, from the questionnaire processing is now out of date and irrelevant. For sure is quite necessary to generate a new survey, in relation with this new situation, generated by the COVID-19 pandemic;
- For sure, there have been changes in people's minds (the new perceptions are totally different than three years ago) and in the management of organizations of any kind (from any important branch); this will affect everyone's activity in relation to the age of 65 (until the age of 65 years and after the age of 65 years);
- EU countries submitted their recovery and resilience plans; these plans are containing measures in terms of working period and retirement age;
- Taking into account all these aspects, I consider the paper to be in no way useful / useful to those interested, because the data of the problem have changed and the options of policy makers have had to be adapted to the new context! Many of the results are no longer relevant so my decision is to reject the paper. I am convinced that the authors can produce a new survey with up-to-date information, so that the new results can directly help the actors involved in these processes.